# Economic Valuation of Green Island, Taiwan: A Choice Experiment Method

**Han-Shen Chen** [1,2,*] and **Chu-Wei Chen** [3]

1   Department of Health Diet and Industry Management, Chung Shan Medical University, No. 110, Sec. 1, Jianguo N. Rd., Taichung City 40201, Taiwan
2   Department of Medical Management, Chung Shan Medical University Hospital, No. 110, Sec. 1, Jianguo N. Rd., Taichung City 40201, Taiwan
3   Department of Applied Economics, National Chung Hsing University, No. 250, Kuo Kuang Rd., Taichung 40227, Taiwan; pat00175@yahoo.com.tw
*   Correspondence: allen975@csmu.edu.tw; Tel.: +886-4-2473-0022 (ext. 12225)

**Abstract:** The evaluation of ecological security and ecosystem services is now a core issue in the field of natural and environmental resources. Quantifying the economic value of island ecosystem services can inform policy decisions that affect the island and help justify expenditures on ecosystem service improvements. This study investigates the preferences of residents and tourists regarding Green Island and estimates willingness-to-pay (WTP) values for island ecosystem services using a choice experiment. The results indicate significant differences between the preferences of residents and tourists regarding island environmental resources. Therefore, based on the multiple attributes and ecosystem services, this study formulated three assessment schemes: "environmental protection", "recreational development", and "integrated operation and management". Based on our analysis of the problems reflected in the aforementioned valuation models, we recommend that policy makers refer to environmental attribute preferences to create statements or advertisements targeting relevant audiences when planning island development. This paper contributes to the literature by demonstrating how the economic valuation of island ecosystem services can help design and target island conservation policies in order to maximize welfare.

**Keywords:** tourism; sustainability; climate change; choice experiment; islands

## 1. Introduction

Increased demands for tourism and recreational resources, as well as a more holistic understanding of sustainability, have led to the rapid development of island tourism. This, in turn, has given rise to research on island development. This, in turn, has given rise to research by scholars who are concerned about island development [1,2]. For example, Scheyvens and Momsen [3] noted that small, developing islands, because of their economic vulnerability, typically use an expanding tourism industry to stimulate economic development [4,5]. Moreover, Li and Yang [6] suggested that coastal overdevelopment, ecological resource destruction, and pollution are unavoidable consequences of island tourism development. Occupational behaviors also arise—such as occupying land for accommodation and water recreational activities—which greatly destabilize the traditional lifestyles of local communities. Since tourism development also creates restrictions with regard to area coverage, natural resources, fragility, disaster recovery capability, and the economy, tourism and recreational services negatively affect the island's environmental, social, and cultural aspects [7–11].

With 1141 km of coastline, Taiwan has abundant marine resources. An emerging tourist destination, island tourism in Taiwan is attracted by ecological, historical, and cultural features.

In particular, Green Island has rich geological and topographical features (coastal terrains, coral reefs); distinctive biological and ecological resources (green sea turtles, flying fish, coconut crabs); and traditional festivals and activities (flying fish festival). Diverse theme-based tours have gradually developed on Green Island, including natural ecology-based tours and relevant experience activities (snorkeling, whale watching, night observation of flying fish). Supported by government policies, island tourism and tribal tourism have become new tourism trends in Taiwan with significant potential for future development. However, tourism growth has also had negative environmental consequences. The construction of coastal embankments can severely damage the coastal environment and destroy biological habitats. Moreover, the introduction of foreign cultures can potentially weaken traditional culture. Given the fragility of island ecosystems, policy makers should develop state land for environmental and cultural protection/conservation while also developing unique ecological and cultural experiences to promote tourism. The development of sightseeing resources must consider sustainable ecological, economic, and social development, while minimizing the impact of recreational activities on the environment. Thus, the adoption of sustainable operating principles and environmental conservation is an important aspect of island tourism development.

Chapin et al. [12] suggested that ecosystem services and biological diversity are important intermediaries between the economic environment and human systems. Further, de Groot et al. [13] argued that in order to value ecosystems, we must first consider ecological, sociocultural, and economic values, and then assess the overall value as a reference for environmental decision-making and management. Barbier et al. [14] estimated the value of ecosystem services such as wetlands, mangroves, coral reefs, seagrass beds, and sandy beaches. Based on changes in land use, Bateman et al. [15] explored the contribution of ecosystem services and ecosystems. Taking four ecological zones in Hangzhou, China, as the research object, Su et al. [16] investigated the effect of landscape pattern and value changes in ecosystem services on urbanization. Maes et al. [17] calculated the value of ecosystem services and used average species richness and species diversity to measure biodiversity.

Due to changes in land-use patterns, the ecosystem functions of natural areas have gradually weakened, and biodiversity has decreased. Ecosystem goods and services provide social benefits such as water supply, recreational activities, and carbon storage [18–20]. Therefore, exploring policies related to changes in land-use patterns, and assessing the benefits of environmental development and protection are important matters of public concern [21]. Given the current situation of Green Island, a complete assessment model for ecosystem services should consider spatial variables such as natural landscape coverage, ecotourism patterns, and land-use patterns. Value functions for assessing ecosystem services should then be constructed and monetized.

Sustainable tourism management is usually conducted in places with high environmental sensitivity, where detailed economic analyses of financial gains vis-à-vis environmental impact are performed prior to developing the area for tourism. Such analyses can support decision making related to the planning, utilization, and sustainable operation of local ecological resources. In the case of Green Island, the conservation efficiency of its eco-environmental resources can only be valued by using non-market goods valuation methods. These methods are classified into two types: revealed preference (RP) and stated preference (SP). In such instances, RP can be used to tease out the values embedded in observed prices. RP directly investigates actual behaviors or results in the target market by using questionnaires. Common RP methods include the traveling cost method (TCM) and the hedonic price method (HPM). Each approach has a different conceptual basis, and can be used to valuate different environmental goods. However, they all share the common feature of using market information or behavior to infer the economic value of an associated non-market impact [22]. TCM is one of the most common methods used in non-market valuation to estimate the recreational values of specific sites [23]. Bertram and Larondelle [24] used TCM to assess the recreational value of forest ruins, while Plant, Rambaldi, and Sipe [25] used HPM to evaluate residents' preferences for tree coverage on urban streets.

The method of stated preference (SP) investigates results that have not yet occurred in the target market to obtain preference data from respondents. SP surveys individual or household preferences and, more specifically, willingness to pay (WTP) for changes in the provision of (non-market) goods, which are related to respondents' underlying preferences. Hence, this technique is of particular value when assessing the effects on non-market goods, the value of which cannot be uncovered by RP methods [22]. SP methods include the contingent valuation method (CVM) and the choice experiment method (CEM). Mark and Swait [26] suggested that SP can overcome certain disadvantages of RP—such as insufficient collinearity between variables and the extent of variation—to make the assessment of parameters more explanatory and clearly reflect respondents' true preferences.

CVM has been widely used to evaluate environmental amenities and damage [27–29]; however, it has several biases. For example, respondents may deliberately conceal their true preferences for non-market goods favoring their personal interests, possibly resulting in strategic biases such as the overestimation or underestimation of value. When a separate inquiry is made for goods, or a mixed inquiry is made for goods, the embedding effect will produce a bias between the values estimated. In the designed questionnaire, the explanatory information and alternative options provided for the study objects are insufficient, creating an information effect. Moreover, when the double-bounded dichotomous choice method is adopted, respondents may ignore the question content because of their own subjective views, and tend to give the same answers to all of the questions, thereby producing an acquiescence bias. The bidding game requires setting a starting price for goods, and that price is used as a benchmark for respondents' comments, which could produce a starting point bias [30].

CEM has become one of the main valuation methods for studying preferences for natural resource conservation. It is also an important preference valuation method for valuating non-market goods [31]. In this regard, Liekens et al. [21] noted that CEM assesses the use and non-use values, defines a hypothetical market by using questionnaires to explore public preferences for landscape conservation and natural development, and further reflects WTP for environmental goods (or services). One widely applicable strategy for valuating these services is to conduct an analysis of all of the various factors determining the output of a good, thereby assessing the contribution of the ecosystem services to the production of that good [23].

The most significant difference between CVM and CEM is that the former can only consider the characteristic attributes of natural resources as a whole commodity for separate value analysis; meanwhile, the latter can distinguish and analyze the multiple attributes of natural resources [32]. As CEM can be used to evaluate multiple attributes and levels, different alternative plans can be combined on the basis of important characteristics associated with non-market goods or services, and choice sets are assumed for different scenarios. In this case, respondents can choose appropriate alternative plans according to their preferences, thus avoiding assessment biases [33]. Therefore, CEM can better solve the problem of comparing profit and loss between the multiple attributes of ecosystem services, and it can reveal public preferences for each eco-functional attribute of ecosystem services [34]. Thus, CEM has been widely used for non-market valuation, including species conservation [35–39], wetland recovery [40–44], ecotourism preferences [45–50], tourists' preferences for land, the environmental functions of national parks [21,39–41,51,52], and the exploration of methods for altering specific ecosystem services to affect economic benefits [34,53–58].

Remoundou et al. [59] employed CEM to evaluate the effects of climate change on WTP for the Santander coastal ecosystem. The study attributes included biodiversity, jellyfish blooms, closed beach days, beach size, and annual household expenditures. Mejía and Brandt [49] used CEM to interview tourists visiting the Galapagos Islands about their WTP for protective measures against invasive species. Their attributes included the depth of experience with the islands' ecosystems, length of stay, level of protective measures taken against invasive species, and price of island tourism. They found that tourists visiting the Galapagos Islands highly valued biodiversity and were marginally willing to pay USD 2543 for better protective measures. Schuhmann et al. [60] employed CEM to evaluate the tourist preferences and WTP for coastal attributes in Barbados. Their attributes included price, type of accommodation,

beach width, distance from beach, and beach litter. Cazabon-Mannette et al. [61] used CEM and CVM to evaluate the non-use value and non-consumptive value of sea turtles in Tobago. Their attributes included price, number of sea turtle sightings, fish diversity (number of species), coral cover, and degree of congestion (number of divers). Xuan et al. [62] used a discrete choice experiment (DCE) to evaluate tourists' WTP for boat tours in the marine-protected area of Vietnam's Nha Trang Bay. Their attributes included coral cover, environmental quality, fishermen's unemployment, and increase in ticket prices. Finally, Peng and Oleson [63] employed DCE to evaluate beach recreationalists' preferences and WTP for improving water quality in Oahu's beaches. Their attributes included water quality, water turbidity, coral cover, fish diversity, and WTP for motor vehicles.

Conditional logit (CL) can be used to estimate the average preferences of tourists from multiple attributes of island tourism and estimate the marginal WTP (MWTP) for these attributes [49,50]. The random parameter logit (RPL) model can reflect the different responses of respondents from different backgrounds toward different attributes. This can be used to examine the heterogeneous preferences of respondents and their WTP for changes in the levels of various attributes (such as folk, cultural, and ecological experiences) [64–66]. To segment a clearer target market, the latent class model (LCM) can segregate respondents into different groups and examine and compare their preferences and group differences (e.g., island tourism preferences, attitudes, and socioeconomic backgrounds) [45]. Based on the aforementioned studies, we can see that the empirical CEM models of CL, RPL, and LCM have been verified for use in the examination and evaluation of multiple attribute preferences for island tourism sites.

These studies show that CEM can be used to construct a multiple attributes utility function for natural resources and the environment to estimate the economic value of goods and services associated with environmental resources. These can include valuation for the conservation and improvement of endangered species populations, service planning and valuation for recreational facilities, and preferences for wetland ecoregion planning. For biodiversity conservation, conservation preferences for different endangered species and the improvement of endangerment levels are the main objectives of species conservation. Establishing a recovery fund system is also an important factor in conservation policies.

In summary, this study assesses the ecosystem service valuation pattern for Green Island. First, the motivation for the study is discussed, and the study objective is proposed. Second, according to the indicators of the ecosystem services of Green Island's ecosystem units, important ecosystem attributes and levels are identified through interviews with relevant experts and researchers, and the attributes and levels of ecological security that were designed into the implementation of the ecological security model are incorporated. Third, CEM is used to construct an ecosystem service assessment utility model. Tourists and residents are categorized into different groups for the questionnaire, and their responses are analyzed to explore differences in WTP for various attributes. Unlike previous studies performed within or outside of Taiwan, this study incorporates the ecological, environmental, and recreational attributes of islands into CEM, and involves the economic benefits of multiple attributes. Finally, based on the results, countermeasures and suggestions are proposed for the sustainable development of the Green Island environment, providing a reference for policy makers to make more efficacious s policies.

## 2. Materials and Methods

### 2.1. Description of Green Island

Green Island, the target area of this study, is located to the southeast of Taiwan island (Figure 1), with a total land area of about 16 km$^2$. Over 3700 people currently live on the island, and its traditional industries are agriculture and fishing. The island has long experienced population decline and aging. Green Island has a great variety of terrain (volcanic island, coastal coral reef); marine ecology; and cultural–historical monuments. In 1995, the Pacific Economic Cooperation Council selected

Green Island as an ecotourism development site, suggesting that Green Island should develop into an eco-resort island. Green Island is gradually transforming its local industrial structure to become more tourism-focused. According to the Tourism Bureau [67], the number of tourists to Green Island increased from 59,383 in 1991 to 345,622 in 2017, which was a growth rate of 482%. Tourism is now an important industry on the island.

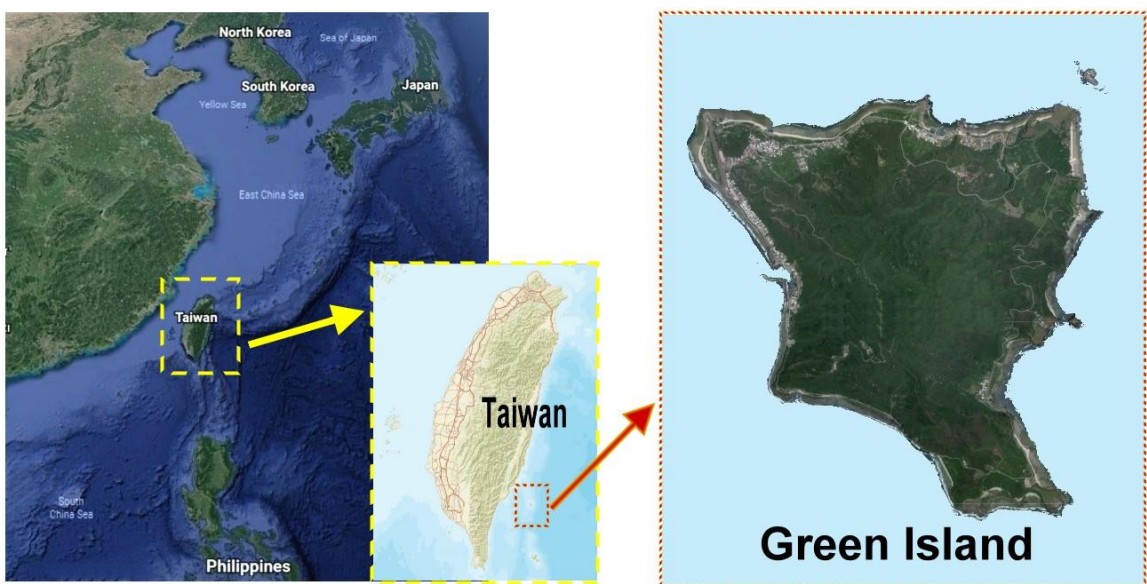

**Figure 1.** Map of the study area.

Due to the fragility of the island ecosystem, offshore development should focus on environmental conservation and cultural preservation, with special ecological and cultural experiences as the main focus. Therefore, the development of tourism resources must consider sustainable ecological, economic, and social development, while reducing the negative impacts of recreation. It is important to ensure that the tourism industry develops under sustainable management and assists ecosystem conservation. For example, Green Island authorities have been implemented to protect environmental resources and reduce resource losses through the development of low-carbon tourism and intertidal zone ecological conservation, along with other measures. However, some conservation plans remain controversial. Therefore, the motivation for this study stems from important environmental protection issues such as biodiversity and the ecological security level.

### 2.2. Construction of a Utility Model for Valuating Ecosystem Services

#### 2.2.1. Multiattribute WTP Valuation Model

CEM was used to construct an ecosystem service assessment utility model for Green Island. An RPL model was used to account for differences between respondents' preferences for attributes in different areas of the island. This model can reflect differences between the preferences of respondents as well as the WTP for different attributes.

CEM is a random utility model that can be used to explore the MWTP for all of the attributes and levels [68]. In the binary model, the utility of the nth respondent is assumed to be determined by the various options the respondent faces ($U_{ni}$), and these options are used to maximize utility, as shown in Equation (1):

$$U_{ni} = V_{ni} + \varepsilon_{ni}, \tag{1}$$

where $U_{ni}$ represents the attribute of the nth respondent facing the ith option, $V_{ni}$ represents the observable part of the utility function, and $\varepsilon_{ni}$ represents the residual item (i.e., the unobservable part).

This study explores differences in preferences and WTP between respondents from different social and economic backgrounds, given various attributes and levels. The analysis was conducted using a random parameters logit (RPL) model. The overall utility of the RPL model is as follows:

$$U_{ni} = V_{ni}(X_{ni}, S_n) + I_{\_ni}, \tag{2}$$

where $V_{ni}$ is the utility coefficient of observable variable $X_{ni}$, and respondent characteristic $S_n$ and represents the respondent's preference.

To estimate the relative importance of all of the attributes of the product in terms of value, it is assumed that the degrees of various attributes in the alternative plan remain the same. Then, the marginal change in WTP for the kth attribute can be given by Equation (3):

$$WTP = \frac{-I2_k}{I2_c}, \tag{3}$$

where $I2_k$ is the parameter on attribute k, and $I2_c$ is the parameter on the payment tool.

2.2.2. Attribute and Level Assessment Design

Utilizing previous reports on Green Island along with conducting interviews with experts from various fields, six attributes were selected: land-use pattern, natural landscape coverage, biodiversity, ecotourism model, ecosystem conservation trust fund, and the ecological security level, which was designed in the implementation result of the abovementioned ecological security model, and was also considered an attribute. Table 1 lists the setting and details of these six attributes.

**Table 1.** Multiple attributes utility assessment indicators for Green Island.

| Attribute | | Level and Current Situation | Ecosystem Service Function | Level Quantity |
|---|---|---|---|---|
| Land-use pattern (LUP) | LUP± LUP+ | 1. Maintain the current situation 2. Increase land use (develop or transform the purposes of the original land) | Supply function: developing traditional agricultural farming patterns and transformation to other land-use purposes | 2 |
| Natural landscape coverage (NLC) | NLC± NLC+ NLC− | 1. Maintain the current situation 2. Increase natural landscape coverage 3. Reduce natural landscape coverage | Support function: providing an environment for different species to inhabit and grow Cultural function: ecotourism and existence value | 3 |
| Biodiversity (BIO) | BIO± BIO+ | 1. Maintain the current situation 2. Increase the species recovery plan | Support function: maintaining biodiversity | 2 |
| Ecotourism model (EM) | EM± EM+ | 1. Maintain the current situation 2. Carry out an environmental education program | Cultural function: providing ecotourism for the public | 2 |
| Ecological security level (ESL) | ESL± ESL+ ESL− | 1. Maintain the current situation 2. Secure status (in which the ecosystem is not affected and there are few ecological disasters) 3. Warning status (in which the ecosystem is damaged to some extent and ecological disasters sometimes occur) | | 3 |
| Ecosystem conservation trust fund (FUND) | FUND | 1. Free of charge (maintain the current situation) 2. NTD250/person/yr [1] 3. NTD500/person/yr 4. NTD1000/person/yr | | 4 |

[1] NTD, new Taiwan dollar (1 NTD = 0.033 USD = 0.028 Euros).

This study constructed a multiple attributes utility assessment model for Green Island, and incorporated the six indicators of the empirical model. The attributes and levels of Green Island were further explored to estimate the MWTP for multiple attributes of Green Island. The results will elucidate the preferences and attitudes of tourists and local residents toward multiple attributes

of Green Island in order to achieve sustainable management goals, such as tourist experience and resource protection.

### 2.3. Preference Selection Combinations for Choice Sets for Green Island Ecosystem Services

After defining the multiattribute utility assessment indicators and the attributes' various levels for the Green Island ecosystem services (Table 1), the choice experiment (CE) evaluation process was used to further describe the preference selection combinations for choice sets to provide a reference for questionnaire and sampling designs. To understand residents' and tourists' multiattribute preferences for Green Island ecosystem services, a more precise improvement plan and the preference for each attribute level needed to be more clearly defined. The arrangement combinations of the various attributes and their levels produced 288 possible factor combinations ($2 \times 3 \times 2 \times 2 \times 3 \times 4 = 288$).

To develop a questionnaire, we used an extensively-used, orthogonal design method (using SPSS), [69]. Using this method, the 288 combinations were reduced to 30 combinations of alternative programs and one status quo alternative. The status quo alternative was included in various choice sets, with each choice including two randomly numbered alternative programs and one status quo alternative. The level of the attribute for the status quo alternative was presented as the current program along with its information. Each questionnaire included the three selected choice sets, and each choice set contained six programs. Thus, there were five versions of the questionnaire. Through the design process and the combination of the aforementioned choice sets, the statistical efficiency of the design of choice sets was improved [45]. Therefore, after deciding on the total number of samples, each respondent randomly selected one questionnaire version for completion.

Each respondent was asked to fill in their answers—that is, select one of the three choice sets (the two alternative programs and one status quo alternative). If the respondent was unable to decide, he or she could select "uncertain;" then, this choice set was considered as a missing value. The various attributes of Green Island ecosystem services and their levels (Table 1), and the content of the choice sets for Green Island preferences (Table 2), were explained to each respondent. This was to help respondents understand the content of the preference attributes of the Green Island ecosystem services before they selected their preferences.

**Table 2.** Example of a choice set for Green Island preferences and programs.

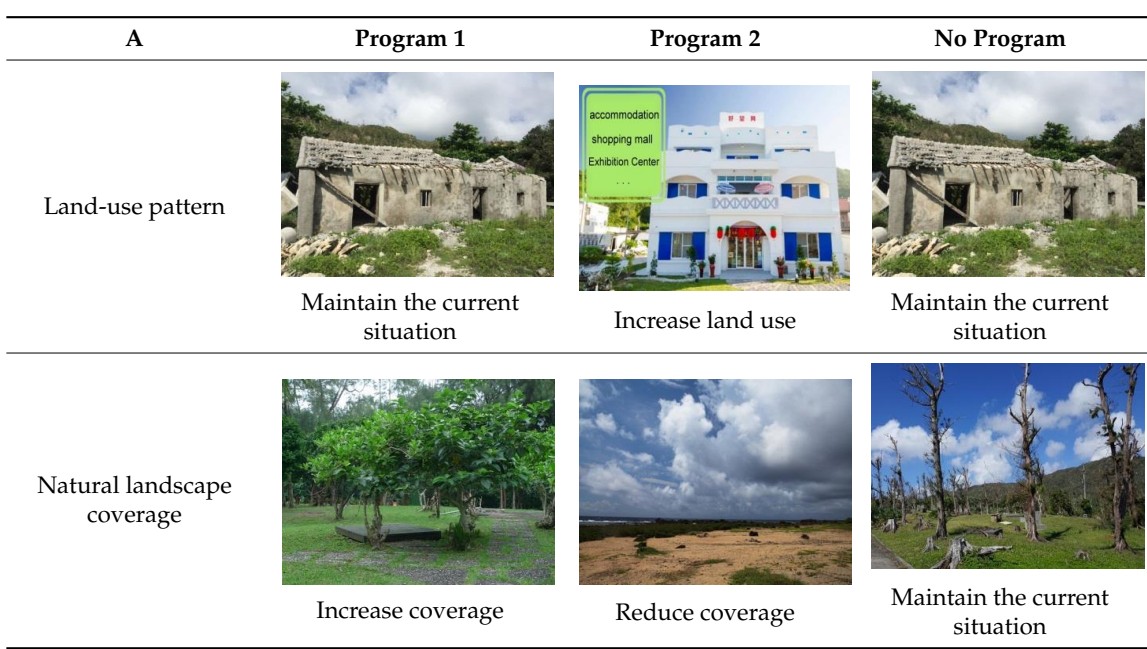

| A | Program 1 | Program 2 | No Program |
|---|---|---|---|
| Land-use pattern | Maintain the current situation | Increase land use | Maintain the current situation |
| Natural landscape coverage | Increase coverage | Reduce coverage | Maintain the current situation |

**Table 2.** *Cont.*

| A | Program 1 | Program 2 | No Program |
|---|---|---|---|
| Biodiversity |  Increase the speciesrecovery plan |  Maintain the current situation |  Maintain the current situation |
| Ecotourism model |  Maintain the current situation |  Carry out an environmental education program |  Maintain the current situation |
| Ecological security level |  Warning status |  Secure status |  Maintain the current situation |
| Ecosystem conservation trust fund | NTD 500/person/yr | NTD 1000/person/yr | Free of charge |
| Which policy do you most prefer? | ☐ | ☐ | ☐ |

*2.4. Survey Design*

Green Island respondents were interviewed in early September 2017 using an initial questionnaire. This questionnaire was then modified based on conservation and management status, expert opinion and advice, and the initial results. Interviews using the final questionnaire were conducted from April to June 2018. A random sampling method and face-to-face interviews were utilized to reduce questionnaire error and help respondents understand the questions. The sampling sites were distributed across the Green Island area, and the respondents were divided into two groups: local residents and tourists. A total of 840 tourists and 420 local residents were interviewed. After incomplete questionnaires were removed, 1021 valid questionnaires remained, representing 81.03% of the total questionnaires. The respondents comprised 653 tourists and 368 local residents (Table 3).

Of the total sample, 517 respondents (50.64%) were female; 577 (56.51%) were single; 444 (43.49%) were married; 276 (42.51%) were aged 30–39 years; 267 (26.15%) were aged 20–29 years; 562 (55.04%) had a university degree; and 281 (27.52%) had a high school degree. Further, 407 (39.86%) earned 20,001–40,000 NTD per month, and 370 (36.24%) earned 40,001–60,000 NTD per month.

**Table 3.** Sociodemographic and economic characteristics of the respondents.

| Description | | Tourists | | Local Residents | |
|---|---|---|---|---|---|
| | | Number | % | Number | % |
| Gender | Male | 317 | 48.55 | 187 | 50.82 |
| | Female | 336 | 51.45 | 181 | 49.18 |
| Marital status | Single | 371 | 56.81 | 206 | 55.98 |
| | Married | 282 | 43.19 | 162 | 44.02 |
| Education | High school | 84 | 12.86 | 197 | 53.53 |
| | University | 418 | 64.01 | 144 | 39.13 |
| | Master's | 151 | 23.12 | 27 | 7.34 |
| Age (yr) | 20–29 | 229 | 35.07 | 38 | 10.33 |
| | 30–39 | 328 | 50.23 | 106 | 28.80 |
| | 40–49 | 74 | 11.33 | 125 | 33.97 |
| | 50–59 | 19 | 2.91 | 81 | 22.01 |
| | $\geq$60 | 3 | 0.46 | 18 | 4.89 |
| Monthly income (NTD) [1] | <20,000 | 121 | 18.53 | 71 | 19.29 |
| | 20,001–40,000 | 214 | 32.77 | 193 | 52.45 |
| | 40,001–60,000 | 282 | 43.19 | 88 | 23.91 |
| | $\geq$60,001 | 36 | 5.51 | 16 | 4.35 |

[1] NTD, new Taiwan dollar (1 NTD = 0.033 USD = 0.028 Euros).

## 3. Results

### 3.1. Analysis of Preferences and Benefits of the Green Island Environmental Resource Attributes

Random parameter logit models were used to analyze tourists' and residents' choice preferences and WTP for Green Island conservation schemes. All of the valuation models passed the goodness-of-fit test and were significantly above the critical value, indicating that the attributes that were selected in this study had sound explanatory capability [70]. Table 4 shows that tourists and local residents had their own preferences for attributes such as "land use", "natural landscape", "biodiversity", "environmental education", and "ecological security level". The RPL model revealed the environmental preferences of each group.

Several factors influenced tourists' environmental preferences (results were significant at the 1% level). (1) The coefficient of land use was negative and significant, indicating that increasing land-use results in a decrease in tourists' utility level; (2) the coefficient of increasing the natural landscape was positive and significant; (3) the coefficients of increasing biodiversity and environmental education were positive and significant, meaning that the species restoration scheme and the increase in the environmental education scheme could raise tourists' preferences for Green Island ecosystem services; and (4) regarding ecological security, the interviewees showed a significant preference for ecological improvement.

The following factors influenced residents' environmental preferences (results significant at the 1% level): (1) The coefficient of land use was positive and significant, indicating that developing or changing land-use patterns could raise the utility level for local residents; (2) the coefficient of increasing the natural landscape coverage was positive, meaning that local residents' utility level could rise with an increase in natural landscape coverage; (3) the coefficients of increasing biodiversity and environmental education were positive, demonstrating that the species restoration scheme and the increase in the environmental education scheme could raise residents' preferences for Green Island's ecosystem services; (4) a relationship was found between improving ecological security and preventing ecological damage, indicating that the utility level of local residents increases with improvements in ecological security.

**Table 4.** Results of the random parameter logit model.

| Variables and Levels | Tourists | | | | | Local Residents | | | | |
|---|---|---|---|---|---|---|---|---|---|---|
| | Coeff. | *t*-Statistic | Coeff. Std | *t*-Statistic | WTP | Coeff. | *t*-Statistic | Coeff. Std | *t*-Statistic | WTP |
| ASC | −0.60686 | −3.70 *** | 1.20282 | 9.49 *** | −2758.45 | 1.27950 | 5.02 *** | 0.08835 | 0.62 | 2611.22 |
| LUP+ | −0.22413 | 5.63 *** | 0.07873 | 1.91 * | −1018.77 | 1.23954 | 15.76 *** | 0.12461 | 2.37 ** | 2529.67 |
| NLC+ | 0.56341 | 8.71 *** | 0.03905 | 0.79 | 2560.95 | 0.80064 | 8.30 *** | 0.09919 | 1.12 | 1633.96 |
| NLC− | −0.47098 | −6.72 *** | 0.02093 | 0.40 | −2140.82 | −0.66028 | −6.59 *** | 0.03658 | 0.34 | −1347.51 |
| BIO+ | 0.47061 | 11.87 *** | 0.01952 | 0.47 | 2139.14 | 0.17376 | 2.58 *** | 0.05111 | 0.79 | 354.61 |
| EM+ | 0.26344 | 6.35 *** | 0.02676 | 0.62 | 1197.45 | 0.24647 | 4.05 *** | 0.03173 | 0.49 | 503 |
| ESL+ | 0.15870 | 2.57 ** | 0.30736 | 4.88 *** | 721.36 | 1.44153 | 11.78 *** | 0.00480 | 0.07 | 2941.9 |
| ESL− | 0.06180 | 0.95 | 0.05081 | 1.10 | − | −0.63198 | −5.74 *** | 0.30378 | 3.05 *** | −1289.76 |
| FUND | −0.00022 | −3.94 *** | − | − | − | −0.00049 | −2.61 *** | − | − | − |
| Number of choice sets | 1959 | | | | | 1104 | | | | |
| Log likelihood ratio | −1664.084 *** | | | | | −775.97 *** | | | | |
| Chi square | 976.20 | | | | | 873.79 | | | | |

* $p < 0.05$; ** $p < 0.01$; *** $p < 0.001$. WTP, willingness to pay; NTD, new Taiwan dollar (1 NTD = 0.033 USD = 0.028 Euros); ASC, alternative specific constant; LUP, land-use pattern; NLC, natural landscape coverage; BIO, biodiversity; EM, ecotourism model; ESL, ecological security level; FUND, ecosystem conservation trust fund.

Tourists' WTP was highest for increasing and maintaining the natural landscape (2561 NTD/person/yr), followed by increasing the species restoration scheme (2139 NTD/person/yr), increasing environmental education (1197 NTD/person/yr), reducing changes in land-use patterns (1018 NTD/person/yr), and healthy ecological levels (721 NTD/person/yr). Tourists were also willing to pay an average of 2140 NTD/person/yr to maintain natural landscapes and prevent resulting damage.

Residents' WTP was highest for upgrading ecosystem security to a healthy level (2942 NTD/person/yr), followed by changes in land-use patterns (2530 NTD/person/yr), increasing natural landscape coverage (1634 NTD/person/yr), increasing environmental education (503 NTD/person/yr), and the species restoration scheme (355 NTD/person/yr). Additionally, local residents were willing to pay an average of 1348 NTD/person/yr for conservation and planning to prevent damage to natural landscapes, and they were also willing to pay an average of 1290 NTD/person/yr for conservation and planning to prevent damage to ecological security.

## 3.2. Difference Analysis of WTP for Resource and Environment Attributes

Cross-analysis of the resource and environment attributes and the respondents' social variables indicated that tourists had differences in preferences for the two attributes of "increasing land use" and "healthy ecological security level"; meanwhile, local residents had differences in the preference for "damaged ecological security level" (Table 4). In other words, the interviewees had different preferences for environmental protection because of their respective positions. Tables 5 and 6 show the resulting correlations between socioeconomic background and WTP when socioeconomic background is taken into consideration regarding interviewees' WTP for the above attributes.

**Table 5.** Relationship between socioeconomic background and WTP for increase in land use.

| Socioeconomic Characteristics | | Tourists | | | Local Residents | | | |
|---|---|---|---|---|---|---|---|---|
| | | Number | Mean WTP (NTD) | *t*-Statistic | F-test | Number | Mean | *t*-Statistic | F-Test |
| Marital status | Single | 371 | −2143.21 | −0.950 | – | 206 | 2541.57 | 0.095 | – |
| | Married | 282 | −2137.76 | | | 162 | 2538.26 | | |
| Education | High school | 84 | −2129.88 | – | 37.670 *** | 197 | 2544.07 | – | 3.279 ** |
| | University | 418 | −2137.71 | | | 144 | 2534.40 | | |
| | Master's | 151 | −2201.15 | | | 27 | 2541.62 | | |
| Age (yr) | 20–29 | 229 | −2136.27 | – | 9.429 *** | 38 | 2539.27 | – | 0.723 |
| | 30–39 | 328 | −2137.26 | | | 106 | 2541.25 | | |
| | 40–49 | 74 | −2144.17 | | | 125 | 2536.76 | | |
| | 50–59 | 19 | −2233.13 | | | 81 | 2542.07 | | |
| | ≥60 | 3 | −2217.28 | | | 18 | 2549.62 | | |
| Monthly income (NTD) | <20,000 | 121 | −2152.91 | – | 2.856 ** | 71 | 2551.57 | – | 5.510 *** |
| | 20,001–40,000 | 214 | −2141.86 | | | 193 | 2538.96 | | |
| | 40,001–60,000 | 282 | −2132.89 | | | 88 | 2531.09 | | |
| | ≥60,001 | 36 | −2156.76 | | | 16 | 2552.79 | | |

** $p < 0.01$; *** $p < 0.001$. WTP, willingness to pay; NTD, new Taiwan dollar (1 NTD = 0.033 USD = 0.028 Euros).

Among tourists aged 50–59, highly educated, high-income tourists preferred to reduce changes in land patterns and land use, whereas highly educated, high-income tourists aged 30–39 preferred improvements in the ecological security level. Local residents with lower educational levels and monthly incomes between 20,000–60,000 NTD preferred changes in land patterns to improve land use. Regarding maintaining the ecological level, local residents with high educational levels and high income more actively supported protecting and maintaining ecological security.

**Table 6.** Relationship between socioeconomic background and WTP for the healthy ecological security level.

| Socioeconomic Characteristic | | Tourists | | | | Local Residents | | | |
|---|---|---|---|---|---|---|---|---|---|
| | | Number | Mean WTP | *t*-Statistic | F-test | Number | Mean WTP | *t*-Statistic | F-Test |
| Marital status | Single | 371 | 514.26 | −0.142 | – | 206 | −1438.60 | −1.196 | – |
| | Married | 282 | 526.19 | | | 162 | −1385.77 | | |
| Education | High school | 84 | 109.57 | – | 9.154 *** | 197 | −1356.24 | – | 11.697 *** |
| | University | 418 | 438.51 | | | 144 | −1431.75 | | |
| | Master's | 151 | 631.00 | | | 27 | −1759.11 | | |
| Age (yr) | 20–29 | 229 | 483.71 | – | 4.850 *** | 38 | −1537.90 | – | 1.558 |
| | 30–39 | 328 | 655.56 | | | 106 | −1446.65 | | |
| | 40–49 | 74 | 381.40 | | | 125 | −1382.14 | | |
| | 50–59 | 19 | 374.31 | | | 81 | −1359.12 | | |
| | ≥60 | 3 | 281.84 | | | 18 | −1455.81 | | |
| Monthly income (NTD) | <20,000 | 121 | 338.46 | – | 2.170 * | 71 | −1359.80 | – | 9.156 *** |
| | 20,001–40,000 | 214 | 505.03 | | | 193 | −1366.72 | | |
| | 40,001–60,000 | 282 | 574.30 | | | 88 | −1481.75 | | |
| | ≥60,001 | 36 | 783.13 | | | 16 | −1883.08 | | |

*$p < 0.05$; *** $p < 0.001$. WTP, willingness to pay; NTD, new Taiwan dollar (1 NTD = 0.033 USD = 0.028 Euros).

## 4. Discussion

The above analysis shows that tourists and local residents have great differences in their preferences for land use. Tourists hope to reduce land use and development, whereas residents hope for changes in land patterns. Further analysis showed that people who prefer increasing land use belong to groups with lower or higher income levels. Presumably, they want to increase their profits by developing and investing in land patterns.

Previous studies have indicated that residents' support for tourism development is affected by local economic conditions, economic benefits for residents, environmental attitudes, and tourism resources [71–73]. Dodds and Holmes [74] pointed out the differences in environmental conservation attitudes between residents and tourists, which varied according to sex, age, educational level. Robledano et al. [75] pointed out that tourists believed that attributes such as natural landscape, biodiversity, and environmental education are important aspects of the lagoon ecosystem. Stefănica and Butnaru [76] argued that responsibility for the environmental impacts of tourism development—which include the destruction of biodiversity, pollution, global warming, increased waste, and natural resource depletion—should be shared by tourism industry operators and tourists alike.

Considering the differences between tourists' and residents' preferences, this study formulated three assessment schemes based on the six attributes and levels of Green Island's ecosystem services (Table 7): the Environmental Protection Scheme, Recreational Development Scheme, and Integrated Operation and Management Scheme. The Environmental Protection Scheme is based on the attributes of increasing the natural landscape coverage area, increasing biodiversity, and maintaining the ecological security level. The Recreational Development Scheme is based on the attributes of increasing the land use, ecotourism mode, and improving the level of ecological security. The Integrated Operation and Management Scheme is based on the environmental protection and recreational guide, combined with the formulation of the social system. This scheme includes four attributes and levels: increasing the natural landscape coverage area, increasing biodiversity, ecotourism mode, and improving the level of ecological security. Our analysis found that the benefit brought by the Environmental Protection Scheme was 7641 NTD/person/yr. The figures for the Recreational Development Scheme and Integrated Operation and Management Scheme were 6684 NTD/person/yr and 8838 NTD/person/yr, respectively.

**Table 7.** WTP for each Green Island ecosystem service management scheme.

| Policy Attributes | Environmental Protection | Recreation Development | Integrated Operation and Management |
|---|---|---|---|
| Land-use pattern | Maintain status quo | Increase land use | Maintain status quo |
| Natural landscape coverage | Increase coverage | Maintain status quo | Increase coverage |
| Biodiversity | Increase biodiversity | Maintain status quo | Increase biodiversity |
| Ecotourism model | Maintain status quo | Implement environmental education | Implement environmental education |
| Ecological security level | Improve security level | Improve security level | Improve security level |
| WTP [1] (NTD/person/yr) | 7641 | 6684 | 8838 |

[1] WTP, willingness to pay; NTD, new Taiwan dollar (1 NTD = 0.033 USD = 0.028 Euros).

It is clear that the best combination for respondents is to increase natural landscape coverage, increase the species recovery plan, conduct environmental education programs, and secure the status of the ecological security level. These results can help inform future management strategies for eco-environmental impact reduction programs on Green Island.

## 5. Conclusions

Sustainable island tourism development requires the integration of recreation, environment, and management information, which is further considered in the decision making for the development and management of sustainable tourism operations. This study used CE to construct a random utility model for Green Island ecosystem services in Taiwan. To do so, it incorporated various factors into the evaluation model for validation, such as recreation (e.g., ecotourism), environment (e.g., land-use pattern, natural landscape coverage, biodiversity, ecological security), and economic considerations (e.g., ecosystem conservation trust fund).

This study found differences between tourists' and local residents' preferences for ecosystem services. Additionally, in terms of the ecological security attribute, tourists and local residents were willing to make improvements and increase maintenance, and people with high educational levels and incomes showed significantly more willingness than others. We further explored the utility function values of attributes and constructed three hypothetical scenarios for future management to analyze respondents' WTP under different management schemes. To provide a more accurate basis for decision making, we suggested that more consideration should be given to local residents' social views and economic factors before policies and measures are promoted by the policy makers [77].

If Green Island were to implement a pricing system, the economic benefits from the aforementioned programs could be combined with the corresponding operation and management costs, and improvements to service packages and measures could be included. This could be used to plan specific content for Green Island tourism development, which, in turn, could be used as a reference for determining the costs of island tourism packages. Lastly, management units and tour operators should seek to understand tourists' preferences and attitudes in order to propose further operation and management strategies that conform to the concept of island tourism and have more specific and feasible market positioning strategies. This would benefit the sustainable development of tourism on Green Island.

This study has several limitations. If the scope of the study can be expanded in the future, the research framework will be more comprehensive. In view of the conclusions and limitations of this study, we put forward the suggestions outlined below.

This study included only set six environmental resource attributes. However, many other attributes could be included, such as recreational facility maintenance, recreational environment maintenance, and recreational population restrictions. In this way, tourists' and local residents' preferences for environmental attributes could be better understood.

The questionnaire was distributed by random sampling. In this study, the socioeconomic backgrounds of the interviewees were examined only with respect to sex, educational level, age, and the data obtained. Our sample had an insufficient number of individuals from different groups to represent the overall perceptions of those groups. Whether different types of interviewees have different preferences for environmental resource protection should be further explored. Other survey items could also be added in future studies, such as attitudes toward environmental protection and type of tourism (e.g., historical sites, natural landscapes, local art).

**Author Contributions:** H.-S.C., the first author, analyzed the data, drafted the manuscript, and acted as corresponding author throughout the submission and revision. C.-W.C. contributed to reviewing and revising the literature.

**Funding:** This work was supported by the Ministry of Science and Technology (Republic of China, Taiwan) (grant number MOST 106-2410-H-040-014). The funder had no role in the study design; in the collection, analysis, and interpretation of data; in the writing of the report; or in the decision to submit the article for publication.

**Conflicts of Interest:** The authors have no conflicts of interest to declare.

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
