# Peer review of "Economic Valuation of Green Island, Taiwan: A Choice Experiment Method"

_sustainability, doi:10.3390/su11020403_

Reviewer 1 Report

Following my recomandations:

- in the introduction I would insert the articulation of the paper

- I would deepen the discussion through references from the scientific literature, highlighting the theoretical implications

- In the conclusions, I would explain the practical implications more clearly. 

Reviewer 2 Report

I have  had  the opportunity  to  read the  manuscript  carefully. The  authors  tried to  explore  how tourists and local residents differently have willingness to pay for ecological attributes. The  article  presents some  interesting  ideas that  are  potentially of  interest  to both  academics  and practitioners.  

However, there  are  some theoretical  and  methodological  issues which  in  my mind  limit  the present  contribution  in its  present  form.

First of all, there are a few  supportive literature  reviews  and theoretical  backgrounds  for this  study.  Although authors  attempt  to cover  some  issues regarding  the  study area,, still  this  article shows  a  lack of  supportive  arguments and  their  consistency and  poor  theoretical development about differences between tourists and local residents, sustainable attributes which were used for variables for research purposes. 

Next,the discussion  doesn't  sufficiently offer  insight  into the  findings.  The author(s)  need  to get  at  question of  "so  what?" more  for  Sustainability.    

Overall, I  enjoyed  reading this  study  and found  it  interesting. It  could  make a  contribution  to sustainable  tourism  study literature  at  micro level.  I  hope my  comments  help in  revising  and rethinking  this  research and  wish  the authors  the  best with  this  line of  scholarship.

Reviewer 3 Report

The paper presents a sound empirical analysis but it lacks clarity and does not develop any theoretical background or any analytical framework. Overall I think the authors can design a publishable paper with the material they have, but in its present form this piece needs a major revision. I hope the following comments help to improve the paper: 

Abstract: please include the real aim of the study. The aim is not the usage of a method, I guess. Furthermore, do not include data in the abstract. Finally, add the contribution (also the theoretical one) of this paper.

Title: I think the paper title is not appropriate and it create expectations which cannot be fulfilled. Sustainable Tourism Development and Climate Change are not the central constructs/objects analyzed in this paper.

The paper structure goes from introduction to methods used: but what about the theoretical background of this study? What about a in-depth literature review? Ecossystem valuation pattern research needs to be discussed. The authors discuss the methods in the introduction section, but the basic theoretical assumptions and its shortcoming and advantages are not discussed or highlighted at all.

Methods and Results: These two section are well-written however very hard to read. The authors might explain the analysis of the data in order to pre-structure the results' section. Furthermore, the study site choice needs justification and the methods used. 

Discussion: A interpretation and discussion of the results in light of earlier studies is totally missing!

Conclusions: this is also a very weak section: it is a repetition what was said or a summary. But it lacks concrete management recommendation as well as further research recommendations. Limitations were addressed.

Reviewer 4 Report

The article focuses on a topic of great interest such as the sustainability of island destinations. The study follows the requirements for a scientific publication. The conclusions set out the main limitations that are basically focused on the attributes and size of the sample in certain groups. However, the methodology can be validated and improved in future studies.

Author Response

Round  2

Reviewer 2 Report

I found that authors tried to incorporate the reviewers’ suggestion, and that this article was neatly revised. 

This article is meaningful in that it deals with economic valuation with CEM for sustainable tourism.It is acceptable for the Sustainability. 

Best,

Reviewer 3 Report

Thank you very much for the detailed and well-explained revision of the document. I think it will be a good contribution to our Journal and I recommend to accept the paper.